## [Peer Review file · Nature Communications]

Prompt Injection Attacks on Vision Language Models in Oncology

Corresponding Author: Professor Jakob Nikolas Kather

Version 0:

Reviewer comments:

Reviewer #1

(Remarks to the Author)

This paper presents the risk of prompt injection attack on vision-language artificial intelligence models in the oncology application. Through experiments the authors demonstrate that embedding sub-visual prompts in medical imaging data can lead the model to produce harmful outputs.

I have several concerns on this work.

1. This position of this work is unclear. This work focuses on prompt injection attack on vision-language artificial intelligence models in the oncology scenario. What are the existing research works on the prompt injection attack on vision-language artificial intelligence models in general scenarios? What are the relatedness and difference between this work and these existing general works? If the prompt injection attack risk of vision-language artificial intelligence models in general scenarios has been revealed, what is the special contribution and novelty of this work?
2. The prompt injection attack setting of this work seems fictional and not real. Who are the attackers and why the attackers can access and modify the user oncology data? It seems that only one that can access the oncology data in this setting is the VLMS service providers. But why they are the attackers? What are the motivations?
3. It seems there is no detailed analysis on why the prompt injection attack on vision-language artificial intelligence models in the oncology application can work, and what are the potential defense methods.

Reviewer #2

(Remarks to the Author)

Summary

The authors investigated how prompt injection attacks affect the performance of large vision-language models (LVLMs) in recognizing lesions in cancer images. Results show that most LVLMs can be misled by the injected prompts, leading to incorrect diagnoses.

Strengths

1. Significant Topic: The investigation of prompt injection attacks in LVLMs is a timely and important issue, particularly in medical diagnosis, highlighting potential risks in clinical settings.
2. Novel Contribution: The study is the first to specifically address prompt injection attacks within the context of medical diagnosis, filling a gap in existing literature.
3. Clear Findings: The results clearly demonstrate that LVLMs can be misled by injected prompts, emphasizing the need for caution in their clinical application.
4. Methodological Soundness: The methodology employed is fundamentally sound, providing a basis for the findings.

Weaknesses

1. Limited Modality Focus: The analysis is primarily restricted to radiology images, with no exploration of other imaging

modalities, which reduces the generalizability of the findings.

2. Expectation of Broader Analysis: Many results align with expected outcomes, suggesting that a more comprehensive investigation into the prompt injection phenomenon is needed.

3. Lack of Solutions: The paper does not propose potential solutions or mitigations for the identified prompt injection vulnerabilities in LVLMs.

4. Insufficient Dataset Details: The specifics of the datasets used are unclear, which is crucial for reproducibility. This lack of detail could introduce biases and make the findings less convincing.

5. Need for Additional Evidence: While the conclusions drawn are reasonable, additional evidence and broader analyses could strengthen the claims made in the study.

Q&A

Q: What are the noteworthy results?

A: Experiments demonstrate that prompt injection attacks can mislead LVLMs, resulting in inaccurate diagnoses.

Q: Will the work be of significance to the field and related fields? How does it compare to the established literature? If the work is not original, please provide relevant references.

A: Yes, the work is significant to the field, as misleading a model to produce a wrong diagnosis could have serious implications. While related works primarily focus on prompt injection attacks in general domains, this study is the first to address this issue in the context of medical diagnosis.

Q: Does the work support the conclusions and claims, or is additional evidence needed?

A: The authors claimed that: "Specifically, we show that embedding sub-visual prompts in medical imaging data can cause the model to provide harmful output", but the analysis only focuses on radiology images and other images modalities are not discussed.

Q: Are there any flaws in the data analysis, interpretation, and conclusions? Do these prohibit publication or require revision?

A: Yes. Although the interpretation and conclusions drawn from the experimental results are reasonable, many findings align with our expectations. Therefore, we anticipate a broader analysis of the prompt injection phenomenon in LVLMs for medical purposes, including additional tasks such as medical report generation and medical visual question answering, as well as exploration of modalities beyond radiology. A wider and deeper analysis would strengthen the publication.

Q: Is the methodology sound? Does the work meet the expected standards in your field?

A: The methodology is sound, but we expect further developments. For instance, do the authors propose any solutions to mitigate prompt injection attacks in LVLMs for medical applications?

Q: Is there enough detail provided in the methods for the work to be reproduced?

A: We appreciate the authors providing some details about prompt designs. However, the specifics of the datasets used are unclear, which is crucial for reproducing the results. Without knowledge of the dataset characteristics—such as modality distribution and dataset size—we cannot assess potential biases in the results, making the experimental findings less convincing.

Reviewer #3

(Remarks to the Author)

This manuscript explores (hidden) prompt injection attacks, on current vision-language models (VLMs). In short, visible medical queries (the official prompt) were found to be modifiable by accompanying hidden & encoded data, which can maliciously alter the VLM output. In particular, three strategies - text prompt injection, visual prompt injection and delayed visual prompt injection (where the visual prompt was hidden in a preceding image) were attempted, with variations as to contrast and text size for visual prompts. A number of variations for each prompt injection attack were attempted, with significant effects on the output observed in general.

While prompt injection is an important issue to be aware of in utilizing VLMs (and LLMs) for medical purposes, some issues might be considered:

1. Firstly, prompt injection is to a certain extent an interface issue (analogous to SQL injection attacks). As such, text prompt injections through hidden/encoded characters can in theory be avoided either by enforcing restrictions on the type of text characters that are allowed through the interface, or through the post-sanitization of input text after submission. As such, while the empirical analysis is thorough and suggests general security vulnerabilities with existing VLMs, this appears to possibly be more of an ephemeral technical issue.

2. Secondly, it is not very clear as to how the text prompt injections were implemented (i.e. through hidden/encoded characters). A search for "text prompt injection" yielded three occurrences in the manuscript, none of which seem to describe how the text prompt injection was implemented. Given that this is a significant detail, it should be thoroughly described.

Version 1:

Reviewer comments:

Reviewer #1

(Remarks to the Author)

Thank the authors for responding to my questions. However, my concerns are not fully addressed.

1. My first concern is "What are the existing research works on the prompt injection attack on vision-language artificial intelligence models in general scenarios? What are the relatedness and difference between this work and these existing general works?" The authors' response is "As of August 24, we did not find a single study on prompt injection attacks on VLMs for medicine on PubMed, arXiv, bioRxiv or medRxiv." In fact, my question is not on whether there is similar work in medicine domain, but in general domain. If there are similar works in general domain, then the finding may be extended to the medical domain, making the novelty of this paper questionable. In fact, I found one similar work in general domain, titled "Empirical Analysis of Large Vision-Language Models against Goal Hijacking via Visual Prompt Injection".
2. My second question is "Who are the attackers and why can the attackers access and modify the user oncology data?" The authors' response is "Usually, the input data is not highly safeguarded. For example, healthcare institutions import imaging data from a variety of external sources" I am not convinced by this claim. If I am a patient and have a medical image generated by a hospital. Why the hospital sends it to a third-party (the attacker) for processing and then get it back for analysis? This scenario is not practical in my opinion.
3. I think the authors' response to my question "It seems there is no detailed analysis on why the prompt injection attack on vision-language artificial intelligence models in the oncology application can work, and what are the potential defense methods" can be more insightful.

Reviewer #2

(Remarks to the Author)

Thank you to the authors for their detailed response, particularly regarding the expanded analysis across multiple modalities, which has addressed several of my concerns. However, I still have a few major questions regarding the revised paper:

1. Justification on the Realism of Prompt Injection Attacks in Medicine: Is prompt injection attack a realistic scenario in the medical field, both currently and in the foreseeable future? While this topic is prominent in discussions about large vision-language models in general, its significance in medical settings raises questions. Given that hospitals prioritize the privacy of their medical data and often utilize private networks for data transfer, the potential for prompt injection attacks to modify medical images and mislead diagnoses seems limited. If a medical image were to be altered during transmission (as noted in Reviewer #1's Comment 2), this would more likely constitute a network security issue rather than a prompt injection attack issue. Both Reviewer #1 and Reviewer #3 have expressed similar concerns.
2. Need for Detailed Solutions: As highlighted in Weakness 3 (and echoed by Reviewer #1), the authors are expected to provide a detailed methodology and corresponding experiments to address the issue of prompt injection attacks, rather than a brief and broad solution as presented in the revised paper.

Reviewer #3

(Remarks to the Author)

We thank the authors for addressing our previous concerns.

Version 2:

Reviewer comments:

Reviewer #1

(Remarks to the Author)

Thank the authors for responding to my questions.

I remain concerned about the realism of prompt injection attacks in the medical field. I agree with Reviewer #2 that this issue seems more like a network security issue than a specific prompt injection attack issue. Additionally, the motivations of external sources, such as outpatient clinics or other hospitals, to carry out a prompt injection attack are not convincing. I believe the authors should provide concrete real-world evidence to demonstrate that this type of attack has actually occurred in the medical domain.

Reviewer #2

(Remarks to the Author)

While authors did not provide solid solutions on preventing prompt injection attack in LLM models, this paper highlights the potential issue, even though the existence of this issue should be consolidated in the real-world setting as concerned by reviewer #1 as well. Given the overall merits of this paper, I am happy to accept it.

Point-by-point response for

“Prompt Injection Attacks on Vision Language Models in Oncology“

Reviewer #1 (Remarks to the Author):

Thank the authors for responding to my questions. However, my concerns are not fully addressed.

1. My first concern is "What are the existing research works on the prompt injection attack on vision-language artificial intelligence models in general scenarios? What are the relatedness and difference between this work and these existing general works?" The authors' response is "As of August 24, we did not find a single study on prompt injection attacks on VLMs for medicine on PubMed, arXiv, bioRxiv or medRxiv." In fact, my question is not on whether there is similar work in medicine domain, but in general domain. If there are similar works in general domain, then the finding may be extended to the medical domain, making the novelty of this paper questionable. In fact, I found one similar work in general domain, titled "Empirical Analysis of Large Vision-Language Models against Goal Hijacking via Visual Prompt Injection".

Response: Thank you for your comment. We acknowledge the existence of studies in the general domain, such as the one you mentioned, "Empirical Analysis of Large Vision-Language Models against Goal Hijacking via Visual Prompt Injection.", which was submitted on arXiv after we submitted the revised manuscript. However, our work is novel in the sense that it applies the concept of prompt injection specifically to the medical domain, which has unique considerations, such as the life-critical nature of the data, the specific ways of using VLMs in oncology, specific tasks and data types, and the healthcare-specific guardrails that VLM providers commonly implement. While findings in general domains can sometimes be extrapolated, the medical context introduces specific challenges and risks that warrant dedicated investigation. E.g., one could assume that prompt injection would only work in non-critical settings, whereas in healthcare, it would be prevented by the alignment and guardrails of the models. Our work shows clearly and for the first time, that this is not the case and that in fact, prompt injection is a relevant threat to the medical domain. In our revised manuscript, we clearly delineate the contributions of our work in relation to existing general studies and emphasize the importance of context-specific research in healthcare.

2. My second question is "Who are the attackers and why can the attackers access and modify the user oncology data?" The authors' response is "Usually, the input data is not highly safeguarded. For example, healthcare institutions import imaging data from a variety of external sources" I am not convinced by this claim. If I am a patient and have a medical image generated by a hospital. Why the hospital sends it to a third-party (the attacker) for processing and then get it back for analysis? This scenario is not practical in my opinion.

Response: We appreciate your concerns about the practicality of this scenario. We are medical doctors with dozens of years of clinical experience. One thing we are doing basically every day is to import

radiology images from CD-ROMs or external servers into our electronic health record. In general, in clinical practice, it is extremely common for medical images to be imported from external sources, such as outpatient clinics or other hospitals. We have visualized the typical pathway of patient data in the following figure, which we have added to the revised manuscript:

The integrity of the data could be compromised during various points in this process, especially if an attacker gains access to the systems involved in image transfer or storage. The prompt injection attack we describe could occur if a) an image is tampered with before it is uploaded to the hospital's system, possibly through compromised external devices or networks, or by patients themselves, or b) at the newly established inference communication with external service providers. We have revised the manuscript to better explain these scenarios and emphasize the realistic attack vectors in the medical field.

3. I think the authors' response to my question "It seems there is no detailed analysis on why the prompt injection attack on vision-language artificial intelligence models in the oncology application can work, and what are the potential defense methods" can be more insightful.

Response: As indicated in several of the publications on prompt injection attacks, building models that are robust against prompt injection attacks is challenging even for state-of-the-art LLM builders like Anthropic, see Hubinger et al., 2024, arXiv (<https://arxiv.org/abs/2401.05566>). As of your suggestion, we addressed this in the revised manuscript in the discussion, as well as an additional set of comprehensive experiments on potential defense methods.

Reviewer #1 (Remarks on code availability):

It would be better if more comments and explanations are provided.

Response: We have improved the code documentation accordingly.

Reviewer #2 (Remarks to the Author):

Thank you to the authors for their detailed response, particularly regarding the expanded analysis across multiple modalities, which has addressed several of my concerns. However, I still have a few major questions regarding the revised paper:

1. Justification on the Realism of Prompt Injection Attacks in Medicine: Is prompt injection attack a realistic scenario in the medical field, both currently and in the foreseeable future? While this topic is prominent in discussions about large vision-language models in general, its significance in medical settings raises questions. Given that hospitals prioritize the privacy of their medical data and often utilize private networks for data transfer, the potential for prompt injection attacks to modify medical images and mislead diagnoses seems limited. If a medical image were to be altered during transmission (as noted in Reviewer #1's Comment 2), this would more likely constitute a network security issue rather than a prompt injection attack issue. Both Reviewer #1 and Reviewer #3 have expressed similar concerns.

Response: Thank you very much for this comment. While hospitals prioritize data privacy, there are manifold scenarios in which data transfer is not secure in healthcare, which all authors (all medical doctors with cumulative dozens of years of experience working in healthcare) have experienced first hand. Further, it is important to recognize that prompt injection attacks and network security threats function as two synergistic concepts for an attack and not as mutually exclusive elements. The prompt injection is the *attack vector*, the network security issue is the *gateway*. Reliance on LLMs (accessed via unsecure browsers as alternatives are scarce) exacerbates network vulnerabilities, making it easier for malicious prompt injections to be inserted. We therefore have to mitigate both vulnerabilities to safeguard medical data.

We have addressed the concern by detailed discussion and an additional explanatory figure in the manuscript.

2. Need for Detailed Solutions: As highlighted in Weakness 3 (and echoed by Reviewer #1), the authors are expected to provide a detailed methodology and corresponding experiments to address the issue of prompt injection attacks, rather than a brief and broad solution as presented in the revised paper.

Response: Thank you for highlighting this. We address this concern by providing compelling evidence on several strategies to prevent prompt-injection on models with fixed weights, with varying levels of success. Alignment training of a VLM is currently out of scope for this manuscript, but from our experiments it becomes clear that this is the most-likely strategy forward (as Claude-3.5 was reportedly focused much more on alignment and shows to be more robust against prompt injection, both in initial experiments and in mitigation efforts. As also mentioned in response to Reviewer 1, preventing prompt injection is not a trivial task as even current SOTA LLM developing companies struggle with this, see Hubinger et al., 2024, arXiv (<https://arxiv.org/abs/2401.05566>). This challenge is precisely why we consider our work highly relevant for both the medical and broader scientific communities, aiming to consolidate mitigation efforts.

Reviewer #3 (Remarks to the Author):

We thank the authors for addressing our previous concerns.

Response: We thank Reviewer 3 for the helpful comments in the first round of revision and for considering our manuscript relevant for publication in its current form.

Point-by-point response for

“Prompt Injection Attacks on Large Language Models in Oncology“

Reviewer #1:

This paper presents the risk of prompt injection attack on vision-language artificial intelligence models in the oncology application. Through experiments the authors demonstrate that embedding sub-visual prompts in medical imaging data can lead the model to produce harmful outputs.

I have several concerns on this work.

1. This position of this work is unclear. This work focuses on prompt injection attack on vision-language artificial intelligence models in the oncology scenario. What are the existing research works on the prompt injection attack on vision-language artificial intelligence models in general scenarios? What are the relatedness and difference between this work and these existing general works? If the prompt injection attack risk of vision-language artificial intelligence models in general scenarios has been revealed, what is the special contribution and novelty of this work?

Author response: Thank you for highlighting this. As of August 24, we did not find a single study on prompt injection attacks on VLMs for medicine on PubMed, arXiv, bioRxiv or medRxiv. The concept is being discussed in several online forums ¹ as well as outside of the medical research community ^{2,3}. Therefore, our study is the very first scientific study of this phenomenon in medicine. We have pointed this out in the revised introduction.

2. The prompt injection attack setting of this work seems fictional and not real. Who are the attackers and why can the attackers access and modify the user oncology data? It seems that only one that can access the oncology data in this setting is the VLMs service providers.

Author response: Thank you for pointing out the lack of discussion on this topic. Our main point is that an attacker does not need access to the VLM at all. An attacker only needs access to the data which is provided as an input to the VLM. Usually, the input data is not highly safeguarded. For example, healthcare institutions import imaging data from a variety of external sources. This is because usually, healthcare data is not considered to be an attack vector. Attacks could also be performed via malicious browser extension or phishing emails, before the input is sent to the VLM service provider. This is a relevant security threat in healthcare scenarios, as real-world evidence shows susceptibility of healthcare personnel to these attack vectors ⁴⁻⁶. Our study fundamentally changes the perception of healthcare cybersecurity. We show for the first time that healthcare data such as images can be used as an attack vector for computer systems involving VLMs. We have clarified this in the revised manuscript.

But why they are the attackers? What are the motivations?

Author response: The attacker that exploits prompt injection could be anyone with malicious intent towards the hospital, e.g. blackmailers, but also an increasing frequency of politically motivated attacks, especially in context of global geopolitical instability^{7,8}. The points raised by the reviewer are excellent, and we have tried to incorporate all of them in the revised manuscript, ensuring that the broader picture of prompt injection attacks is better presented.

3. It seems there is no detailed analysis on why the prompt injection attack on vision-language artificial intelligence models in the oncology application can work, and what are the potential defense methods.

Author response: The general concept of prompt injection attacks exploits the fundamental goal of the VLM, which is to complete the user's prompt and provide a helpful response. As the user's initial prompt and the attack vector would enter the model simultaneously, the model would treat the attack vector as part of the user's prompt and therefore comply with it exactly as it would with every user instruction. Therefore, designing VLMs to act as helpful assistants is what makes them susceptible to these attacks. Possible defense methods would be alignment training of the VLM or deploying additional models to safeguard the main VLM. Our work points out a prompt-based vulnerability of healthcare VLMs for the first time and therefore provides a strong rationale for the further development and deployment of such mitigation measures. We have added these points to the revised discussion section of our manuscript.

Reviewer #2:

Summary

The authors investigated how prompt injection attacks affect the performance of large vision-language models (LVLMs) in recognizing lesions in cancer images. Results show that most LVLMs can be misled by the injected prompts, leading to incorrect diagnoses.

Strengths

1. Significant Topic: The investigation of prompt injection attacks in LVLMs is a timely and important issue, particularly in medical diagnosis, highlighting potential risks in clinical settings.
2. Novel Contribution: The study is the first to specifically address prompt injection attacks within the context of medical diagnosis, filling a gap in existing literature.
3. Clear Findings: The results clearly demonstrate that LVLMs can be misled by injected prompts, emphasizing the need for caution in their clinical application.
4. Methodological Soundness: The methodology employed is fundamentally sound, providing a basis for the findings.

Author response: Thank you very much for pointing out the novelty and significance of our work.

Weaknesses

1. Limited Modality Focus: The analysis is primarily restricted to radiology images, with no exploration of other imaging modalities, which reduces the generalizability of the findings.

Author response: Thank you for raising this excellent point. We have addressed this by extending our study to multiple additional imaging modalities in which visual cancer diagnosis is essential: endoscopic imaging, histopathology and clinical photographs of melanomas, doubling our initial sample size to a total of 594 attacks. We are confident that this increases the statistical power as well as the generalizability of our findings.

2. Expectation of Broader Analysis: Many results align with expected outcomes, suggesting that a more comprehensive investigation into the prompt injection phenomenon is needed.

Author response: We agree that the results align with the expected outcome that prompt injection works. However, our study is the first to systematically and scientifically show and quantify this effect. We think that by doubling our sample size, including a broader range of modalities, and more granular analysis per modality, we have thoroughly addressed your request for a more comprehensive investigation.

3. Lack of Solutions: The paper does not propose potential solutions or mitigations for the identified prompt injection vulnerabilities in LVLMs.

Author response: Thank you for highlighting this aspect, which was also asked for by Reviewer 1. While we cannot provide a single solution for this issue that we deem a fundamental design problem of LLMs, we have now included an outlook on (hybrid) alignment training, a new method via which ethical goals can be balanced with human preferences. Our study provides scientific evidence for prompt injection vulnerabilities in clinical VLMs for the first time and is therefore a clear motivation for further investigation and deployment of mitigation strategies. We have addressed this point in the revised discussion section.

4. Insufficient Dataset Details: The specifics of the datasets used are unclear, which is crucial for reproducibility. This lack of detail could introduce biases and make the findings less convincing.

Author response: We have addressed this request by adding all raw data, including radiologist's report for all locally obtained cases in Supplementary Table 2. All cases obtained from publicly available resources are listed with corresponding reference, descriptions for these cases are based on information from the original resource, and labeled diagnosis were checked for plausibility by board-certified experts for radiology, dermatology and internal medicine, respectively. This makes our study fully reproducible by anyone.

5. Need for Additional Evidence: While the conclusions drawn are reasonable, additional evidence and broader analyses could strengthen the claims made in the study.

Author response: Thank you for mentioning this. Upon your and other reviewers' suggestion, we have broadened our analysis to three additional imaging modalities, namely endoscopy, histology and photography. The new results are in line with our original findings and further strengthen our conclusions.

Q&A

Q: What are the noteworthy results?

A: Experiments demonstrate that prompt injection attacks can mislead LVLMs, resulting in inaccurate diagnoses.

Q: Will the work be of significance to the field and related fields? How does it compare to the established literature? If the work is not original, please provide relevant references.

A: Yes, the work is significant to the field, as misleading a model to produce a wrong diagnosis could have serious implications. While related works primarily focus on prompt injection attacks in general domains, this study is the first to address this issue in the context of medical diagnosis.

Q: Does the work support the conclusions and claims, or is additional evidence needed?

A: The authors claimed that: "Specifically, we show that embedding sub-visual prompts in medical imaging data can cause the model to provide harmful output", but the analysis only focuses on radiology images and other images modalities are not discussed.

Author response: Thank you very much. Addressing your comment, we extensively improved our manuscript, doubling the sample size and amount of imaging modalities. Thereby, our study provides substantial evidence for vulnerabilities of medical VLMs across a broad range of image modalities.

Q: Are there any flaws in the data analysis, interpretation, and conclusions? Do these prohibit publication or require revision?

A: Yes. Although the interpretation and conclusions drawn from the experimental results are reasonable, many findings align with our expectations. Therefore, we anticipate a broader analysis of the prompt injection phenomenon in LVLMs for medical purposes, including additional tasks such as medical report generation and medical visual question answering, as well as exploration of modalities beyond radiology. A wider and deeper analysis would strengthen the publication.

Author response: As requested, we have substantially broadened our analysis by doubling the sample size and adding multiple additional image modalities. Also, as suggested by the reviewer, we investigate the task of generating structured reports from images. Further tasks, such as visual medical question answering, or incorporation of VLMs in multi-component AI workflows, is currently out of scope, but will be the subject of our future studies. We have added this point to the revised discussion section.

Q: Is the methodology sound? Does the work meet the expected standards in your field?

A: The methodology is sound, but we expect further developments. For instance, do the authors propose any solutions to mitigate prompt injection attacks in LVLMs for medical applications?

Author response: Thank you for highlighting this. We agree that the initial version lacked proposed solutions. Based on our refined statistical analysis, as well as renewed literature research we have now proposed possible solutions to mitigate harm by prompt injection attacks, including hybrid alignment training and agent systems. We have added this to the revised discussion section.

Q: Is there enough detail provided in the methods for the work to be reproduced?

A: We appreciate the authors providing some details about prompt designs. However, the specifics of the datasets used are unclear, which is crucial for reproducing the results. Without knowledge of the dataset characteristics—such as modality distribution and dataset size—we cannot assess potential biases in the results, making the experimental findings less convincing.

Author response: We have updated the Supplementary Data to include the original image report for all locally obtained images, as well as related data. Our work is now fully reproducible by anyone. All images and corresponding diagnosis were double-checked by our board-certified clinicians for dermatology, pathology, radiology and internal medicine, as now described in more detail in the methods.

Reviewer #3:

This manuscript explores (hidden) prompt injection attacks, on current vision-language models (VLMs). In short, visible medical queries (the official prompt) were found to be modifiable by accompanying hidden & encoded data, which can maliciously alter the VLM output. In particular, three strategies - text prompt injection, visual prompt injection and delayed visual prompt injection (where the visual prompt was hidden in a preceding image) were attempted, with variations as to contrast and text size for visual prompts. A number of variations for each prompt injection attack were attempted, with significant effects on the output observed in general.

Author response: Thank you very much for this excellent summary.

While prompt injection is an important issue to be aware of in utilizing VLMs (and LLMs) for medical purposes, some issues might be considered:

1. Firstly, prompt injection is to a certain extent an interface issue (analogous to SQL injection attacks). As such, text prompt injections through hidden/encoded characters can in theory be avoided either by enforcing restrictions on the type of text characters that are allowed through the interface, or through the post-sanitization of input text after submission. As such, while the empirical analysis is thorough and suggests general security vulnerabilities with existing VLMs, this appears to possibly be more of an ephemeral technical issue.

Author response: Thank you for this interesting analogy. The difference between SQL injection attacks and visual prompt injection attacks is that the input data (which is usually not safeguarded) is used as the attack vector in prompt injection attacks. Trivial filters and safeguards of user interfaces can prevent SQL injection attacks, but it is much more difficult, if not impossible, to shield healthcare IT systems from visual prompt injection attacks. In the revised manuscript, we suggest several options to harden healthcare IT systems against visual prompt injection. We suggest (a) better alignment training, potentially as hybrid alignment training between ethical outputs and human-preferred outputs, and (b) an embedding of the VLM in agent-based solutions, potentially with a “post-sanitization” process as suggested by reviewer. The solutions to this are not trivial, and should be implemented along the implementation of LLMs/VLMs in healthcare settings in general. Our study provides a strong motivation to implement such mitigation measures in real-world IT systems.

2. Secondly, it is not very clear as to how the text prompt injections were implemented (i.e. through hidden/encoded characters). A search for "text prompt injection" yielded three occurrences in the manuscript, none of which seem to describe how the text prompt injection was implemented. Given that this is a significant detail, it should be thoroughly described.

Author response: Thank you for highlighting this flaw in our initial manuscript submission. We have now described this essential aspect more clearly in our methods section. To investigate the effect of text prompt injection, it was not necessary to hide the text in the prompt, as the threat of invisible characters as sources of adversarial attacks are a well-investigated subject⁹. Therefore, we consider the text prompt injection arm of our study as a type of positive control for prompt injection since it was unaltered. In contrast, we explored sub-visual injections as more harmful examples of prompt injections that are not detectable by humans.

References for point-by-point response

1. Timbrell, D. The Beginner's Guide to Visual Prompt Injections: Invisibility Cloaks, Cannibalistic Adverts, and Robot Women. <https://www.lakera.ai/blog/visual-prompt-injections>.
2. Samoilenko, R. Prompt injection attack on ChatGPT steals chat data. *System Weakness* <https://systemweakness.com/new-prompt-injection-attack-on-chatgpt-web-version-ef717492c5c2> (2023).
3. Liu, Y. *et al.* Prompt Injection attack against LLM-integrated Applications. *arXiv [cs.CR]* (2023).
4. Gordon, W. J. *et al.* Assessment of employee susceptibility to phishing attacks at US health care institutions. *JAMA Netw. Open* **2**, e190393 (2019).
5. Jalali, M. S., Bruckes, M., Westmattelmann, D. & Schewe, G. Why employees (still) click on phishing links: Investigation in hospitals. *J. Med. Internet Res.* **22**, e16775 (2020).
6. Cartwright, A. J. The elephant in the room: cybersecurity in healthcare. *J. Clin. Monit. Comput.* **37**, 1123–1132 (2023).
7. Bruce, M., Lusthaus, J., Kashyap, R., Phair, N. & Varese, F. Mapping the global geography of cybercrime with the World Cybercrime Index. *PLoS One* **19**, e0297312 (2024).
8. Law, R. Cyberattacks on healthcare: Russia's tool for mass disruption. *Medical Device Network* <https://www.medicaldevice-network.com/features/cyberattacks-on-healthcare-russias-tool-for-mass-disruption/> (2024).
9. Boucher, N. & Anderson, R. Trojan Source: Invisible Vulnerabilities. in *32nd USENIX Security Symposium (USENIX Security 23)* (USENIX Association, Anaheim, CA, 2023).

REVIEWERS' COMMENTS

Reviewer #1 (Remarks to the Author):

Thank the authors for responding to my questions.

I remain concerned about the realism of prompt injection attacks in the medical field. I agree with Reviewer #2 that this issue seems more like a network security issue than a specific prompt injection attack issue. Additionally, the motivations of external sources, such as outpatient clinics or other hospitals, to carry out a prompt injection attack are not convincing. I believe the authors should provide concrete real-world evidence to demonstrate that this type of attack has actually occurred in the medical domain.

Response: Thank you for your comment. As we pointed out in the added illustration (Figure 1b) and in the response to reviewer 2's comments, network security issues and prompt injection attacks are not mutually exclusive, but two concepts that work together very well. Network security issues answer the question of HOW to get into the system, prompt injection attacks are a one (new) possibility of WHAT to do once in the system. Hence, an attacker could easily use both. However, while network security issues have been described extensively, as we acknowledge in the manuscript, prompt injection attacks are a completely new type of threat that will only become relevant for healthcare with the emergence of VLM-use cases in medicine. As the first vision-language models for medicine have been published only in this year, approval as medical devices (and therefore usage in real healthcare settings) is pending^{1,2}. For example, PathChat, which was originally published in Nature earlier this year¹, has since led to the founding of Modella AI. The company's current product, PathChat 2, is now seeking approval as a medical device³. While we therefore cannot provide real world examples of prompt injection attacks that have already happened, our study provides the first-ever report of the relevance of this concept for medicine. In parallel, Y Yang et al. have provided evidence for a similar attack form with poisoned training data⁴, which we now reference.

Remarks on code availability:

The quality has improved since last version.

Response: Thank you for acknowledging this.

Reviewer #2 (Remarks to the Author):

While authors did not provide solid solutions on preventing prompt injection attack in LLM models, this paper highlights the potential issue, even though the existence of this issue should be consolidated in the real-world setting as concerned by reviewer #1 as well. Given the overall merits of this paper, I am happy to accept it.

Response: Thank you for your comment. As we show in the newly created Figure 4, enforcing ethical outputs can, to some extent and depending on the model, mitigate prompt injection attacks. We acknowledge that many more studies will be necessary on this crucial aspect.

To our knowledge, prompt injection attacks have not occurred in a real-world setting with potential patient harm. However, this is most likely not a reflection of whether or not prompt injections attacks are relevant for healthcare. Rather, it is a consequence of the current status of implementation of VLMs into medicine. These tools, as any other, must undergo rigorous approval processes as medical devices, a process that is currently performed, e.g. for Modella AI, which was founded to bring the VLM "PathChat"

to clinical practice.^{1,3} We therefore argue that this makes our study even more relevant at this point in time. Our report on the concept of prompt injection attacks could put the spotlight on this security-relevant issue, enforcing model developers to implement mitigation techniques already for first-ever approval.

References

1. Lu, M. Y. *et al.* A Multimodal Generative AI Copilot for Human Pathology. *Nature* (2024)
doi:10.1038/s41586-024-07618-3.
2. Zhang, K. *et al.* A generalist vision-language foundation model for diverse biomedical tasks. *Nat. Med.* 1–13 (2024).
3. Modella AI. <https://www.modella.ai/index.html>.
4. Yang, Y., Jin, Q., Huang, F. & Lu, Z. Adversarial attacks on Large Language Models in medicine. *arXiv [cs.AI]* (2024).